# How do supervisor-student relationship affect anxiety of graduate students? ——The mediating role of research self-efficacy and the moderating role of mindset

Peng Li[1], Xiaoqing Tang[1], Yongjie Sun[2], Songjie Huang [3]*

**1** Institute of vocational education, Tongji University, Shanghai, China, **2** School of psychology and cognitive science, East China Normal University, Shanghai, China, **3** School of education, Tsinghua University, Beijing, China

* jessieh1114@163.com

## Abstract

### Objective

In China, the number of graduate students continues to increase, and graduate education is developing rapidly. However, mental health issues among graduate students are widespread. Self-efficacy and mindset have been found to moderate the effect of negative life events on individuals' anxiety and depression levels. However, few studies have extended this conclusion to postgraduate student samples.

### Methods

This study utilized a moderated mediation model to explore the mediating role of research self-efficacy and the moderating role of mindsets. A sample of Chinese graduate students ($n = 2278$) was included in the study.

### Results

The results indicated that the supervisor-student relationship significantly affects the anxiety level of graduate students. Satisfaction with the supervisor-student relationship was negatively associated with anxiety. Furthermore, research self-efficacy was found to significantly mediate the impact of supervisor-student relationships on anxiety. Additionally, mindset significantly moderated the association between supervisor-student relationships and both anxiety as well as research self-efficacy. For students with a growth mindset, both their research self-efficacy and anxiety levels were less influenced by the quality of the supervisor-student relationship.

**Data availability statement:** Data supporting this study are available in the Open Science Framework (OSF) repository at https://osf.io/sgp5t/ (DOI: https://doi.org/10.17605/OSF.IO/SGP5T).

**Funding:** The author(s) received no specific funding for this work.

**Competing interests:** The authors have declared that no competing interests exist.

## Conclusions

These findings highlight the critical importance of healthy supervisor-student relationships for graduate students' anxiety and depression, and suggest that fostering a growth mindset can mitigate the negative effects of poor supervisor-student relationships on mental health. Therefore, efforts should be made to cultivate students' growth mindsets to enhance their mental well-being. In addition, this study extends the application of self-determination theory to the field of education by verifying the mediating role of research self-efficacy.

## Introduction

As high-quality talent resources for social development, graduate students play a significant role in promoting China's economic development. However, the mental health issues among graduate students are prominent. A survey by Nature showed that 39% of graduate students experienced depression, and the tendency for depression and anxiety in this group was six times higher than that of the general population [1]. A survey conducted by Nature in 2019 indicated that 40% of Chinese PhD students sought help for depression and anxiety during their studies, compared to 36% in other countries [2]. This demonstrated that Chinese graduate students generally endure multiple intertwined pressures from academic, employment, finances, interpersonal relationships, and emotional distress, leading to varying degrees of mental health problems. Among these, the supervisor-student relationship significantly impacts the emotions and lives of graduate students [3].

China's graduate education adopts a supervisor-responsible system, where supervisors are crucial in the overall development of graduate students. The relationship between graduate students and the supervisors affects the quality of graduate education. However, in reality, some supervisor-student relationships are still less than ideal, with conflicts in values, academic guidance, research outcomes, moral cognition, moral behavior, and academic ideals versus real life [4]. For instance, the shocking case of Chinese PhD student Qi Tailei shooting his advisor in August 2023 highlighted these issues. Based on this, this study targeted master's and doctoral students to investigate the relationship between supervisor-student relationships and anxiety, and to explore the roles of research self-efficacy and mindsets, thereby providing suggestions for improving the mental health levels of graduate students.

### Supervisor-student relationship and anxiety

The supervisor-student relationship is a dynamic where graduate students engage in coursework, scientific research, and thesis writing under the guidance of the supervisor, learning both professional and personal skills in the process [5]. Essentially, it is a mutually beneficial and cooperative relationship between supervisors and students [6,7]. This relationship, centered on educational interaction, is inherently complex.

The quality of the supervisor-student relationship has profound implications, particularly on the mental health of students [8]. Liang et al. [3] explored the impact of teacher-student relationships on students' subjective well-being from the perspective of reciprocal teacher-student relationships. Using longitudinal data for analysis, the researchers found that reciprocal teacher-student relationships were positively correlated with students' subjective well-being, and that the trust level between supervisors and students had a significant moderating effect. Chu et al. [9]conducted survey at three universities in Beijing, China. The results of 448 Chinese graduate students showed that student-supervisor relationship quality was negatively correlated with graduate students' problematic Internet use. However, problematic Internet use is a behavioral indicator, while anxiety and depression are indicators of psychological states. Therefore, research is still needed on the relationship between student-supervisor relationship quality and anxiety and depression. The graduate students' satisfaction with their supervisors was also found to be significantly associated with their levels of depression [10]. Specifically, oppressive behavior by supervisors can lead to increased anxiety or depression among students. PhD students with strained relationships with their supervisors are more anxious compared to those more released with the supervisors [11,12]. Thus, advisors are "significant others" to graduate students, meaning they are individuals or groups that have a substantial impact on the students' personal development [13]. Accordingly, we proposed Hypothesis 1: The supervisor-student relationship has a significant impact on graduate students' anxiety. A good mentoring relationship can alleviate students' anxiety, while a poor mentoring relationship exacerbates it.

## The mediating role of research self-efficacy

Research self-efficacy is an application and development of Bandura's self-efficacy concept in the field of scientific research. It refers to the degree of confidence that individuals engaged in research activities have in their ability to use their capabilities to complete specific research tasks [14,15]. Students' perceptions of the supervisor-student relationship are positively correlated with their research self-efficacy [16]. Specifically, the more guidance students receive from their supervisors, the higher their research self-efficacy. Supervisors' support, trust, respect, and granting of research autonomy to students during the research process can help boost students' confidence in their own abilities and creativity [17]. Conversely, if the student-supervisor relationship is unsatisfactory, it can greatly undermine graduate students' confidence in completing research tasks [18]. According to Self-Determination Theory, individuals have three basic psychological needs: competence, relatedness, and autonomy [19]. The extent to which these needs are met can reflect an individual's mental health and well-being, and the need for competence is synonymous with self-efficacy [20]. Projecting this to the academic and research domain, it can be concluded that research self-efficacy is negatively correlated with anxiety. Moreover, Ma et al. [21] found in a survey of 1,095 Chinese postgraduate students that both instrumental and emotional support from supervisors was negatively correlated with postgraduate students' anxiety, and that research self-efficacy partially mediated the relationship. The current study will analyze a larger range of data, verify this relationship, and improve the generalizability of the conclusions. A study of 325 doctoral students at a medical university also reached a similar conclusion, finding that research self-efficacy was significantly negatively correlated with depression and anxiety [22]. Based on the dynamics that graduate students' doubts about their own abilities and loss of confidence in their own research can increase their anxiety and depressive tendencies, we proposed Hypothesis 2: Research self-efficacy mediated between the mentoring relationship and graduate students' anxiety.

## The moderating role of mindset

The theory of growth mindset and fixed mindset was first proposed by Carol [23]. Mindset refers to the implicit theories about whether human traits, characteristics, emotions, and experiences are fixed or malleable. According to the theory proposed by Carol [24], an individual's mindset lies on a continuum from fixed to growth. People with a more fixed mindset believe that individual traits are stable and unchanging, while those with a growth mindset believe that

human traits are malleable [23], while a more growth mindset influences people's attributions and achievement goal orientations, which further affect their motivational beliefs such as self-efficacy [25,26]. Schroder et al. [27] used longitudinal data to analyze the relationship between individuals' mindset towards anxiety and their pain and anxiety. It was found that fixed mindset towards anxiety can positively predict pain and anxiety in the next five weeks, while individuals with growth thinking towards anxiety have an average reduction of anxiety and pain of one-third of SD after five weeks. A substantial body of research has demonstrated that growth mindset positively predicts self-efficacy among entrepreneurs, teachers' instructional efficacy, and self-efficacy in creative games [28]. Furthermore, mindset has been shown to significantly predict students' academic achievement [29,30]. As a moderating factor, mindset can moderate the relationship between cyberbullying and depressive symptoms in adolescents [31]. Specifically, adolescents with a growth mindset are less affected by cyberbullying and exhibit fewer depressive symptoms compared to those with a fixed mindset. Thus, mindset affects how people view adversity and challenges, which in turn influences their emotional and behavioral functioning. Individuals with a growth mindset are less likely to be negatively affected by adverse experiences, and growth mindset interventions can reduce students' psychological distress [31]. Chang et al. [32] analyzed the relationship between the family socioeconomic status and depressive symptoms of 1,572 Chinese children and found that growth mindset significantly moderated the relationship between the family socioeconomic status (SES) and depression symptoms of children, and the moderating effect was more significant for children with lower SES. Therefore, this study proposes hypothesis:

H3: Mindset can significantly moderate the impact of the supervisor-student relationship on research self-efficacy.

H4: Mindset can significantly moderate the influence of the supervisor-student relationship on anxiety.

## Goal of the present study

Based on the above assumptions and hypotheses, this research introduced a moderated mediation model (See Fig 1) that examined the mechanism of the supervisor-student relationship on anxiety. Furthermore, we aimed to examine the mediating role of research self-efficacy and the moderating role of mindset.

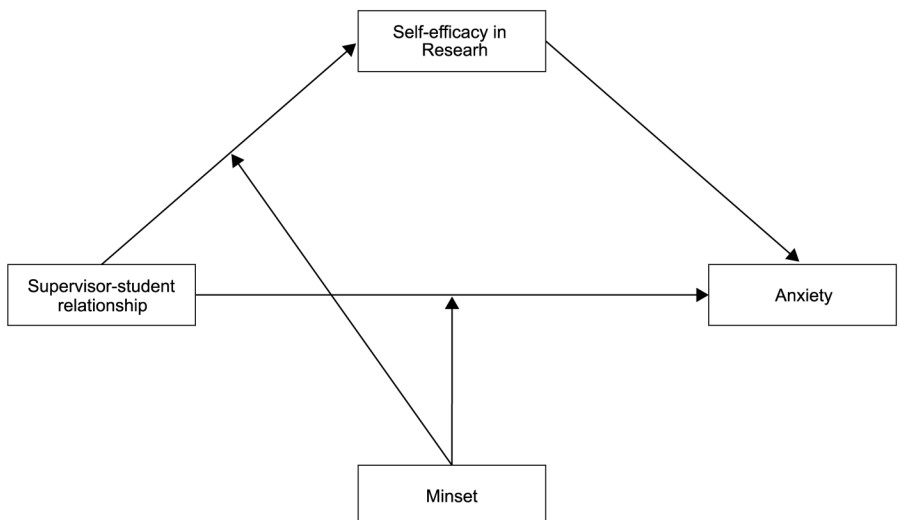

**Fig 1. Diagram of hypotheses of the moderated mediation model.**

## Method

This study was approved by the IRB of Tongji University (No. tjdxsr2024044). This study adhered to the principle of minimal risk. Prior to the formal commencement of the research, and in accordance with the Helsinki Declaration, this study required participants to provide informed consent. Participants could take part in the study after confirming that they have thoroughly read, understood, and agreed to the informed consent items pertaining to this research. Throughout the study, participants were free to withdraw at any time without incurring any penalty or negative consequences.

### Participants and settings

A total of 2278 participants (including 1114[49.0%] males and 1335[58.6%] master students) were recruited online in this study. The questionnaire was collected using the Wenjuanxing platform, participants were invited to complete the survey through electronic links distributed via email and social media. Responses that were incomplete or exhibited patterns of response bias (e.g., completion times under 5 minutes or over 10 minutes) were excluded from analysis. Only fully completed questionnaires with valid data were retained, and all responses recorded were used for analysis. Responses were collected anonymously to reduce common method bias (Podsakoff et al., 2003). Participants were required to be graduate student and aged over 18.

### Measures

The measurement tools in this study included the Mentoring Functions Questionnaire-9 (MFQ-9), Research Self-Efficacy Scale (RSES), Depression Anxiety Stress Scale (DASS), and Mindset Scale. The Cronbach's α coefficients for the MFQ-9, RSES, DASS (anxiety subscale), and Mindset Scale were 0.910, 0.914, 0.876, and 0.930, respectively, indicating acceptable reliability. Given that participants were native Chinese speakers, the questionnaires were translated into Chinses using the back-translation method suggested by Brislin [33]. Furthermore, a pilot study was conducted to test the validity of the translated questionnaire.

### Mentoring functions questionnaire-9 (MFQ-9)

Scandura [34] first proposed the MFQ scale, which was refined by Castro & Williams [35] into a 9-item scale. Castro & Williams [35]
found the reliability of the MFQ-9 assessment to be 0.91. In addition, Hu et al. [36] also used the MFQ-9 to measure supervisor-student relationships, and in their sample of Taiwanese students, the reliability of this scale was 0.93.The questionnaire employs a 5-point Likert scale, with scores ranging from one ("completely disagree") to five ("completely agree"). A higher score indicates greater student satisfaction.

### Research Self-Efficacy Scale (RSES)

Three dimensions are included in the RSES Designed by Gess et al., [37]: research methods and communication, norms and organization, and research communication. The scale employs a five-point Likert Scale, with a higher score indicating stronger research self-efficacy.

### Depression Anxiety Stress Scale (DASS)

The anxiety subscale from the DASS designed by Clark and Watson [38] was also used in this study. It assesses the degree of the negative emotional states on a five-point Likert scale, with a higher score indicating a higher level of anxiety.

### Mindset scale

Designed by Dweck [39], the mindset scale was rated on a five-point Likert scale. Midkiff et al. [40] analyzed the mindset scale using Item Response Theory (IRT), with a Cronbach's α = 0.93. A higher score indicates growth mindset, while lower score suggests fixed mindset.

## Data analysis

**Analyses plan.** First, a pilot study was conducted to test the validity of the questionnaire and it was analyzed using Amos 24. The mediating model was also tested using Amos 24, and for the moderated mediation model, PROCESS macro of SPSS was used with Model eight, with the Bootstrap method applied to certificate the significance of the effects. A simple slope analysis was conducted to test the direction of the moderation effect. Gender was included as a covariate in all tests of the models to reduce the impact of additional factors. Data are centered before model testing.

**Pilot testing.** Before conducting the formal survey, 325 graduate students from universities in Shanghai, China, were recruited in a pilot study for testing the validation of the questionnaire. The results showed that the $KMO = 0.939$, and the chi-square statistic of Bartlett's test of sphericity was 30494.029, $p < 0.001$, indicating the questionnaire's suitability for factor analysis. Confirmatory factor analysis using AMOS 24 showed an overall acceptable result, with all the factor loadings > 0.7. See Tables 1 and 2 for details.

# Results

## Common method bias tests

The data in this study were derived from participants' self-reports, which might lead to common method bias. Harman's single-factor test was used to assess common method bias. The results showed that there were five factors with eigenvalues exceeding one, and the variance explained by the first factor was 30.375%, indicating that common method bias did not significantly impact the study's results.

## Description analyses and correlation analyses for the variables

Table 3 shows the description and correlation analyses on the variables. A series of t-tests were conducted to compare the difference between the master students and doctoral students on the variables. The results showed that in terms of

**Table 1. Results of the Factor Analyses on Each Scale and the Overall Questionnaire.**

|  | $\chi^2/df$ | RMSEA | CFI | NFI | TLI | RFI | IFI | CR | AVE |
|---|---|---|---|---|---|---|---|---|---|
| MFQ-9 | 2.109 | 0.022 | 0.998 | 0.997 | 0.997 | 0.994 | 0.998 | 0.895 | 0.549 |
| RSES | 2.115 | 0.022 | 0.998 | 0.997 | 0.997 | 0.995 | 0.998 | 0.833 | 0.699 |
| DASS | 1.520 | 0.015 | 0.999 | 0.998 | 0.999 | 0.996 | 0.999 | 0.810 | 0.525 |
| Mindset Scale | 1.317 | 0.018 | 0.996 | 0.997 | 0.998 | 0.993 | 0.996 | 0.765 | 0.577 |
| Overall | 1.934 | 0.020 | 0.990 | 0.979 | 0.988 | 0.976 | 0.990 | 0.842 | 0.593 |

*Note.* RMSEA: Root-mean-square error of approximation, ranging from 0–1, with a lower value indicating better fitness; CFI: Comparative fit index, ranging from 0–1, with a higher value indicating better fitness; NFI: Normed fit index, ranging from 0–1, with a higher value indicating better fitness; TLI: Tucker-lewis index, ranging from 0–1, with a higher value indicating better fitness; RFI: Relative fit index, ranging from 0–1, with a higher value indicating better fitness; IFI: Incremental fit index, ranging from 0–1, with a higher value indicating better fitness; CR: Composite reliability, a CR > 0.7 indicates a good convergent validity; AVE: Average variance extracted, an AVE > 0.5 indicates a good convergent validity. The above indicators are based on Steiger [41].

**Table 2. Results of the Heterotrait-Monotrait Ratio (HTMT) between scales.**

|  | MFQ-9 | RSES | DASS |
|---|---|---|---|
| MFQ-9 |  |  |  |
| RSES | 0.354 |  |  |
| DASS | 0.510 | 0.478 |  |
| Mindset Scale | 0.466 | 0.634 | 0.457 |

**Table 3. Description Analyses and Correlations between the variables.**

| Variables | | Master Student | | Doctoral Student | | 1 | 2 | 3 | 4 |
|---|---|---|---|---|---|---|---|---|---|
| | | M | SD | M | SD | | | | |
| 1 | Supervisor-student relationship | 4.02 | 0.77 | 3.74 | 0.87 | 1 | | | |
| 2 | Research self-efficacy | 4.00 | 0.84 | 3.71 | 0.93 | 0.442** | 1 | | |
| 3 | Anxiety | 2.25 | 0.82 | 2.40 | 0.88 | −0.402** | −0.417** | 1 | |
| 4 | Mindset | 3.65 | 0.61 | 3.51 | 0.62 | 0.219** | 0.268** | −0.206** | 1 |

Note. **$p < 0.01$.

supervisor-student relationship, that of master students was significantly better compared to doctoral students, $t(2189)$ = 7.32, $p < 0.001$, Cohen's $d = 0.321$, $95\%CI[0.234, 0.407]$. Similar differences were also found on the anxiety, $t(2189)$ = −3.93, $p < 0.001$, Cohen's $d = -0.172$, $95\%CI[-0.258, -0.086]$, as well as on the research self-efficacy, $t(2189) = 7.18$, $p < 0.001$, Cohen's $d = 0.314$, $95\%CI[0.228, 0.401]$. In terms of the mindset, a significantly higher score on master students indicated a growth mindset compared to doctoral students, $t(2189) = 5.413$, $p < 0.001$, Cohen's $d = 0.237$, $95\%CI[0.151, 0.323]$.

Furthermore, a correlation analysis was conducted on the variables. The results illustrated significantly positive correlation between supervisor-student relationship and research self-efficacy, which had a significantly negative correlation with anxiety. The direct effect between supervisor-student relationship and anxiety was significantly negative.

### The mediating role of self-efficacy in research between supervisor-student relationship and anxiety of graduate student

The results of mediation model showed an acceptable fit index, $\chi^2/df = 2.207$, $RMSEA = 0.023$, $CFI = 0.992$, $NFI = 0.985$, $TLI = 0.990$, $RFI = 0.983$, $IFI = 0.992$. The bias-corrected non-parametric percentile Bootstrap method recommended by Hayes [42] was adopted for the mediation effect test, with 5,000 resamples to calculate 95% confidence intervals. Results showed that the total effect ($\beta = -0.4133$, $p < 0.001$, $95\%CI[-0.451, -0.374]$), direct ($\beta = -0.278$, $p < 0.001$, $95\%CI[-0.319, -0.237]$) and indirect effects ($\beta = -0.135$, $95\%CI[-0.162, -0.110]$) were significant. Specifically, supervisor-student relationship was positively associated with the research self-efficacy, $\beta = 0.476$, $p < 0.001$, $95\%CI[0.437, 0.516]$, while it was significantly negative correlated with the anxiety, $\beta = -0.278$, $p < 0.001$, $95\%CI[-0.319, -0.237]$. In terms of the indirect effect, a similar negative correlation was found between the research self-efficacy and anxiety, $\beta = -0.283$, $p < 0.001$, $95\%CI[-0.321, -0.245]$. See Fig 2 for the path relationship among the variables.

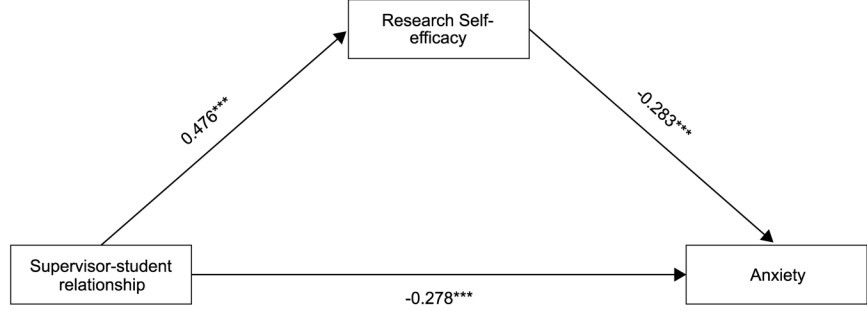

**Fig 2. Path coefficient result of the mediation model.**

### The role of mindset in research between supervisor-student relationship and anxiety of graduate student

To test the role of Mindset, this study employed Model eight in PROCESS macro program of SPSS designed by Hayes [43]. Research self-efficacy and anxiety were involved as dependent variable, with supervisor-student relationship as well as the interaction between supervisor-student relationship and mindset involved as predicted variable. Results showed that the interaction effect between the supervisor-student relationship and mindset was significant in predicting research self-efficacy, β=−0.081, $p<0.001$, 95%CI[−0.142, −0.021], as well as in predicting anxiety, β=0.079, $p<0.01$, 95%CI[0.021, 0.136]. See Table 4 and Fig 3 for details.

A simple slope test was conducted to test the direction of the effect of the mindset. Those scoring below $M$-1$SD$ on the mindset were classified as fixed mindset and those above $M+1SD$ were classified as growth mindset. In terms of the association between the supervisor-student relationship and research self-efficacy, compared to fixed mindset (β=.470), those with a growth mindset (β=.368) had a significantly weaker effect on the dependent variable, $p<.001$, 95%CI[.306,.431], indicating a significant moderation effect between the variables. Similarly, the negative association between supervisor-student relationship and anxiety was significantly fewer when moderated by growth mindset (β=−.207) compared to fixed mindset (β=−.304), $p<.001$, 95%CI[.268, −.146]. See Fig 4 for details.

**Table 4. Tests for Moderation Effect of Mindset.**

| Outcome Variable | Predicted Variable | β | t | 95%CI |
|---|---|---|---|---|
| Self-efficacy in research | Supervisor-student relationship | 0.419 | 19.845*** | [0.378, 0.460] |
| | Mindset | 0.253 | 9.382*** | [0.200, 0.306] |
| | Supervisor-student relationship * Mindset | −0.081 | −2.640** | [-0.142, -0.021] |
| | $R^2$ | 0.229 | | |
| | F | | 168.411*** | |
| Anxiety | Supervisor-student relationship | −0.255 | −11.819*** | [-0.298, -0.213] |
| | Mindset | −0.097 | −3.740*** | [-0.148, -0.046] |
| | Supervisor-student relationship * Mindset | 0.079 | 2.693** | [0.021, 0.136] |
| | $R^2$ | 0.240 | | |
| | F | | 143.546*** | |

Note. **$p<0.01$, ***$p<0.001$.

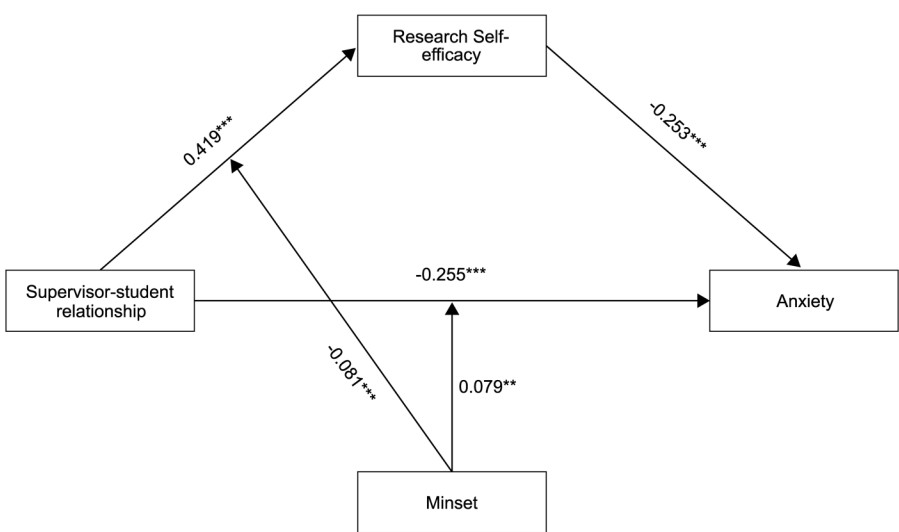

**Fig 3. Path coefficient result of the moderated mediation model.**

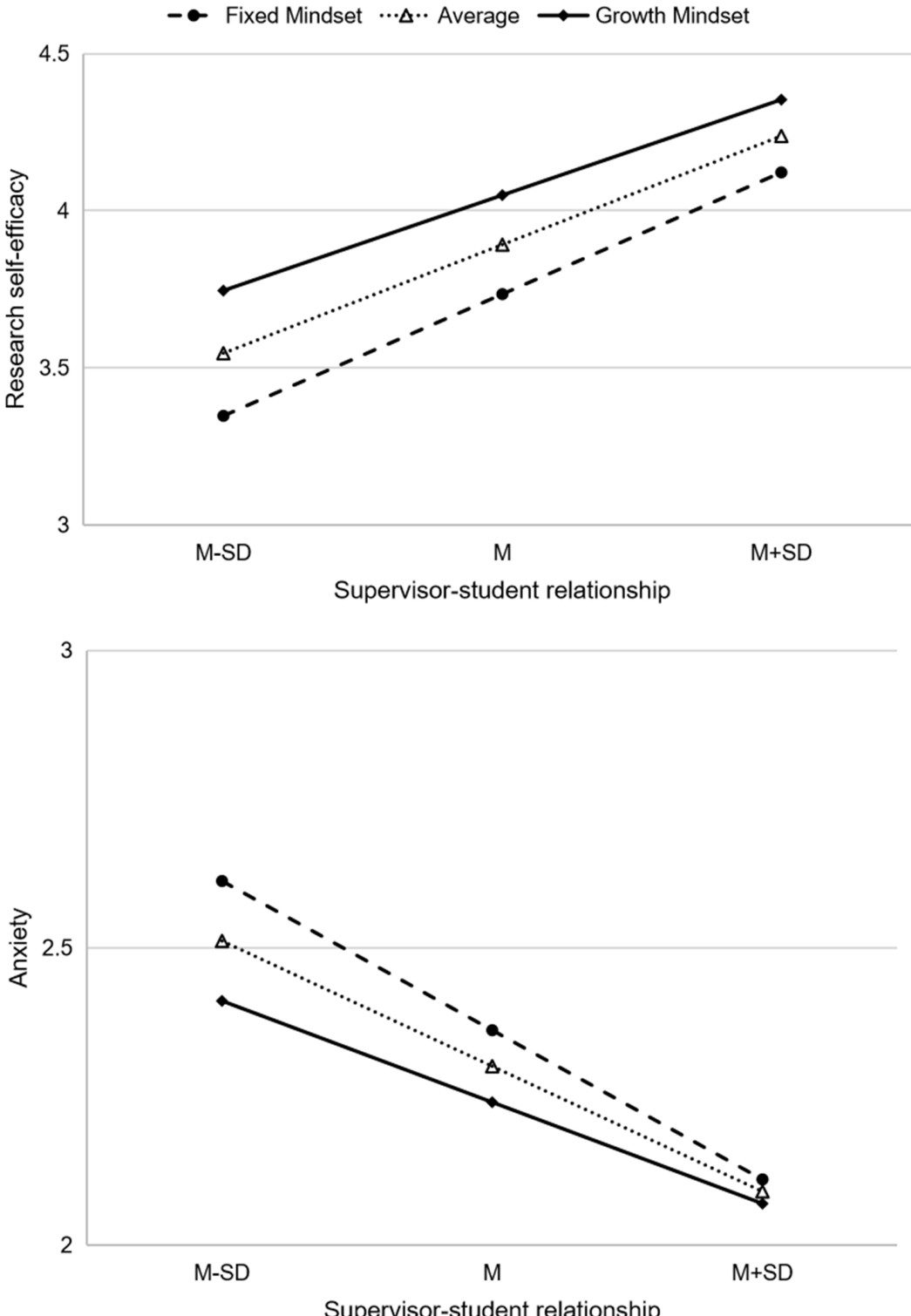

**Fig 4. Simple slope tests of mindset.**

## Discussion

Graduate students face pressures from various aspects such as academics and employment, which might lead to anxiety. As a key figure in the graduate stage, supervisors can have a profound impact on graduate students' mental health. In the current study, a moderated mediation model was constructed to examine the relationship between the supervisor-student relationship and anxiety as well as research self-efficacy in a group of graduate students in China. Furthermore, the moderating role of mindset was testes. It reveals the mechanism by which the supervisor-student relationship influences anxiety through research self-efficacy, and this mechanism is moderated by mindset.

First, the results showed that a significantly negative association was found between the supervisor-student relationship and anxiety. This indicated that a good supervisor-student relationship alleviates students' anxiety, while a poor relationship exacerbates it. Scholars have generally noted that besides the students themselves, supervisors are often the core figures influencing graduate students' learning trajectories and subjective well-being [11,12,44]. This further proves that supervisors are "significant others" to students. Furthermore, the findings of this study corroborate the conclusions of the study based on Chinese sample revealed that one's subjective well-being was a key factor that affected the anxiety and depression [45]: the supervisor-student relationship can negatively influence students' anxiety levels. China's educational culture, deeply rooted in Confucian traditions, emphasizes reverence for teachers and scholarly authority. Cross-cultural studies reveal significant variations in how "respect for educators" manifests across societies[e.g. 46,47]. In Western individualistic contexts (e.g., the United States), student-teacher relationships are typically grounded in reciprocal equality, characterized by what scholars term "horizontal respect" [48]. Conversely, Confucian-influenced societies (e.g., Taiwan) maintain hierarchical academic relationships where students in junior positions were expected to demonstrate deferential respect, a distinction identified as the most pronounced cross-cultural variation in educational interactions [49]. This cultural framework amplifies the psychological significance of supervisor-student relationships for Chinese graduate students. Empirical evidence suggests that relationship quality exerts greater impact on students in Confucian contexts compared to their Western counterparts, with Chinese students more likely to internalize supervisors' evaluations as reflections of personal competence. A survey of the supervisor-student relationship was conducted among students in China and the UK by Spencer-Oatey [49]. The results showed that Chinese students felt a greater power gap with their mentors, thereby impacting their research self-efficacy. This pattern was also found in the following analyses in the current study. The results of the mediation model showed that research self-efficacy significantly mediated the association between the supervisor-student relationship and anxiety. The concept of self-efficacy was initially proposed by Bandura [14,37] and later extended to research self-efficacy, which refers to an individual's degree of confidence in solving problems in their current research [22,37]. A substantial body of existing research has indicated that higher self-efficacy is significantly associated with lower levels of negative psychological states such as depression and anxiety [50,51]. The findings of this study also corroborate this point, demonstrating that research self-efficacy has good criterion validity among the scientific research field. In a good supervisor-student relationship, supervisors provide support to graduate students, which enhances their confidence in completing research tasks. A poor supervisor-student relationship leads to doubts about the students' abilities and reduces their confidence in completing research tasks, making it difficult for them to smoothly carry out research work. On the other hand, research have found that an individual's self-efficacy is a key factor determining their decision-making and behavior [52,53]. The current study found that research self-efficacy could influence graduate students' mental health. A low research self-efficacy would lead students being fear in conducting academic research, which led to anxiety. While a high research self-efficacy makes students confident in producing academic results and are motivated to conduct academic research, thus alleviating anxiety.

Secondly, the interaction effect of the supervisor-student relationship and mindset on research self-efficacy was found to be significant. Simple slope analysis shows that the negative effect of the supervisor-student relationship on research self-efficacy was stronger for graduate students with a fixed mindset than for students with a growth mindset. This

indicated that the more fixed the mindset, the greater the influence of the supervisor-student relationship on research self-efficacy. If the supervisor-student relationship deteriorates, graduate students with a fixed mindset are more likely to lower their research self-efficacy, while those with a growth mindset are more likely to maintain higher research self-efficacy. Individuals with a growth mindset believe that their research abilities are controllable and changeable [54]. Regardless of whether they encounter a good or poor supervisor-student relationship, they always have confidence in their research abilities. A similar significant interaction effect of the supervisor-student relationship and mindset on anxiety was also found, indicating that the more fixed the mindset, the greater the impact of the supervisor-student relationship on anxiety. A growth mindset can help individuals cope with stressful life events and adopt more adaptive emotion regulation strategies [25,54]. Students with a growth mindset believe that anxiety is controllable, and there is a positive correlation between a growth mindset and reduced psychological distress [55]. The impact of a growth mindset on mental health operates through cognitive mechanisms. The first mechanism is causal explanation or attribution when facing negative life events [56,57]. Graduate students with a growth mindset do not entirely attribute a poor supervisor-student relationship to their own reasons and do not blame themselves. The second cognitive mechanism emphasizes the inferential consequences of negative life events [56,58]. People with a growth mindset believe that experiencing stress is a form of training. Therefore, graduate students with a growth mindset view a poor supervisor-student relationship as an opportunity for development, thus reducing psychological distress.

## Implications of the study

The findings of this study provide practical implications for enhancing the mental health of Chinese graduate students. First, universities should systematically establish a mentorship quality monitoring system by integrating supervisor-student relationship management into supervisor training programs. Evidence-based courses on mentoring strategies should be developed, as existing research demonstrates that supervisors' instrumental support (e.g., academic guidance) and emotional support positively correlate with students' subjective well-being [21]. Therefore, cultivating supervisors' emotional intelligence and autonomy-granting techniques should be prioritized. Second, graduate training programs need to incorporate psychological capital development components. Grounded in Self-Determination Theory – which identifies autonomy, competence, and relatedness as three fundamental psychological needs – future interventions should design self-efficacy enhancement modules through structured training featuring phased goal achievement and peer mentoring exchanges. Third, universities should innovate mindset intervention models by embedding growth mindset cultivation into academic supervision. Aronson et al. [59] demonstrated that simple interventions like lectures and video observations significantly improve students' growth mindset, a finding corroborated by Andersen & Nielsen's [60] study showing educational pamphlets and videos effectively enhance children's malleable thinking. Implementing these systemic measures requires coordinated efforts among academic affairs departments, faculty teams, and student support services, supported by continuous outcome evaluation to establish dynamic optimization mechanisms, ultimately forming an ecosystem conducive to graduate students' mental health development.

## Limitations and future recommendations

This study employed a questionnaire survey to collect data from graduate students across different academic levels in China. However, the research has several limitations: First, while cross-sectional data analysis revealed significant correlations among the key variables, the cross-sectional design precludes causal inference [61,62]. To explore the potential causal mechanisms underlying the relationship between supervisor-student relationships and anxiety/depression symptoms, future studies should adopt longitudinal designs with multi-wave data collection for predictive analysis. Second, although anonymous questionnaire administration was implemented to mitigate social desirability bias, self-reported measurements may still entail common method bias. The dual-control strategies proposed by

Podsakoff et al. [63] offer solutions: procedurally through psychologically separated questionnaire designs (e.g., interleaving items from distinct research topics), and statistically via Multitrait-Multimethod (MTMM) models to disentangle method variance. Third, while focusing on generalized anxiety, existing evidence demonstrates significant associations between self-efficacy and specific anxiety subtypes such as social anxiety [64,65]. Future investigations should differentiate anxiety manifestations in graduate populations (e.g., academic anxiety, career development anxiety) to develop targeted interventions. Additionally, incorporating validated psychological mediators like subjective well-being and resilience would help systematically construct theoretical models of how supervisory support influences graduate mental health.

## Conclusion

This study examined the impact of supervisor-student relationships on graduate students' anxiety, exploring the mediating role of research self-efficacy and the moderating role of mindset. Results supported the hypothesized moderated mediation model. Positive supervisor-student relationships were associated with lower anxiety levels, partially mediated by research self-efficacy. Mindset moderated the effects of supervisor-student relationships on both anxiety and research self-efficacy, with a growth mindset mitigating the negative impacts of poor relationships. These findings highlight the importance of fostering healthy supervisor-student relationships and suggest that cultivating a growth mindset may help alleviate the negative effects of poor relationships on mental health. Future research and practical interventions should focus on improving supervisor-student interactions, enhancing research self-efficacy, and fostering growth mindsets to promote graduate students' mental well-being and academic success.

## Author contributions

**Conceptualization:** Peng Li, Xiaoqing Tang, Songjie Huang.

**Data curation:** Peng Li.

**Formal analysis:** Peng Li, Xiaoqing Tang, Songjie Huang.

**Investigation:** Peng Li, Songjie Huang.

**Methodology:** Xiaoqing Tang, Songjie Huang.

**Resources:** Yongjie Sun, Songjie Huang.

**Supervision:** Peng Li.

**Validation:** Xiaoqing Tang, Yongjie Sun.

**Visualization:** Xiaoqing Tang.

**Writing – original draft:** Yongjie Sun, Songjie Huang.

**Writing – review & editing:** Yongjie Sun, Songjie Huang.

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
