## [Decision Letter · Decision Letter 0]

21 Mar 2025

PONE-D-25-05351How do Supervisor-Student Relationship Affect Anxiety of Graduate Students?——The Mediating Role of Research Self-efficacy and the Moderating role of MindsetPLOS ONE?

Dear Dr. Huang,

Thank you for submitting your manuscript to PLOS ONE. After careful consideration, we feel that it has merit but does not fully meet PLOS ONE’s publication criteria as it currently stands. Therefore, we invite you to submit a revised version of the manuscript that addresses the points raised during the review process.

We look forward to receiving your revised manuscript.

Kind regards,

Naeem Mubarak, PhD

Academic Editor

PLOS ONE

Journal Requirements:

Additional Editor Comments:

The manuscript has a good deal of merit for publication after major revisions. The recommended citation  is optional and should only be taken as a suggestion for the improvement of the manuscript.  

Reviewers' comments:

Reviewer's Responses to Questions

**Comments to the Author**

1. Is the manuscript technically sound, and do the data support the conclusions?

Reviewer #1: Partly

2. Has the statistical analysis been performed appropriately and rigorously?

Reviewer #1: Yes

3. Have the authors made all data underlying the findings in their manuscript fully available?

Reviewer #1: No

4. Is the manuscript presented in an intelligible fashion and written in standard English?

Reviewer #1: Yes

Reviewer #1: I am pleased to have the opportunity to review your manuscript. The research topic is meaningful, but the manuscript currently has numerous issues that need to be addressed. In its current state, it is not sufficient for publication in this journal. Below are my detailed comments and suggestions for improvement:

Abstract

Q1: The abstract lacks a clear statement of the research gap.

Q2: The conclusion should include both theoretical and practical implications of all findings (e.g., the significance of Research Self-efficacy).

Q3: In conclusion, the author expanded the scope of the research results. Anxiety is only one aspect of mental health, and this study is a cross-sectional quantitative study. The measurement of "anxiety" and "mental health" differs in scope and tools. Therefore, the conclusions should be carefully revised to align with the specific dependent variable (anxiety) studied.

Introduction

Q4: While the authors clearly explain why this study was conducted, the introduction lacks an analysis of the research gap and the study’s innovation.

Q5: Before discussing the relationships between variables, the author lacks the reason and basis for studying the mediator and moderator. Specifically, what is the necessity of studying "Research Self-efficacy" and "Mindset"?

Q6: The literature review should not merely list studies or evidence but should critically analyze the literature. Without this, the innovation and necessity of the study are questionable.

Q7: The literature review on the "The moderating role of mindset" has critical flaws, leading to logical confusion. First, the current literature reasoning does not sufficiently support H3 and H4; the reasoning suggests that "mindset" might function more as a mediator. Second, "mindset" and "growth mindset" are distinct concepts and variables. The authors should confirm and unify the variables and hypotheses (ensuring consistency in the title, hypotheses, and measurement tools). I recommend referencing high-quality core literature in the field as a logical paradigm.

Q8: The literature review lacks sufficient support from recent studies (within the last five years).

Research Methods

Q9: Please provide a detailed description of the sampling process. For example, how were the questionnaires distributed? How was bias controlled during the distribution process? What was the number of valid questionnaire? What were the inclusion and exclusion criteria for valid questionnaire? Was the minimum sample size sufficient, and what is the theoretical basis for it?

For guidance on bias control, please refer to: Podsakoff, P. M., MacKenzie, S. B., Lee, J. Y., & Podsakoff, N. P. (2003). Common method biases in behavioral research: a critical review of the literature and recommended remedies. Journal of Applied Psychology, 88(5), 879. https://doi.org/10.1037/0021-9010.88.5.879 (This is optional should only be taken as a suggestion for the improvement of the manuscript). 

Q10: Regarding the measurement tools, I have concerns about the "Mentoring Functions Questionnaire-9 (MFQ-9)" and the "Mindset Scale." Can these tools accurately measure "supervisor-student relationship" and "growth mindset"? Additionally, what is the reliability of these tools?

Q11: The section on ethical approval should be moved before the description of the “Participants and settings”.

Results

Q12: How many questionnaires were used for the confirmatory factor analysis (CFA)? Note that, methodologically, CFA should not use the same data as exploratory factor analysis (i.e., pilot study).

Q13: In addition to model fit indices (Table 1), what are the other structural validity indicators (e.g., convergent validity, discriminant validity)?

Q14: Please provide literature support for the standard value ranges of all indicators.

Q15: Validity tests could be moved to the methods section and integrated with the measurement tools. The common method bias test should be the first part of the results section and should be moved accordingly.

Q16: Since the accuracy of two scales in measuring the study variables is debatable, the validity of the results is questionable.

Discussion

Q17: The discussion should focus on analyzing the reasons behind the findings rather than merely restating the results. The discussion would benefit from more recent literature, especially when analyzing reasons, and should consider the Chinese cultural context.

Q18: What are the practical implications of this study? The authors should propose actionable recommendations based on the findings.

Limitations

Q19: Please suggest specific solutions to address the study’s limitations rather than general statements. For example, what objective measurement methods could future studies use to assess variables like anxiety?

In summary, the manuscript requires substantial revisions before it can be considered for publication. I look forward to reviewing an improved version of your work.

**Do you want your identity to be public for this peer review?** For information about this choice, including consent withdrawal, please see our Privacy Policy

Reviewer #1: No

---

## [Author Response · Author response to Decision Letter 1]

26 May 2025

Dear Editors and Reviewers:

We appreciate the opportunity to revise our manuscript and are grateful for the insightful comments provided by the reviewers. Those comments are all valuable and very helpful for revising and improving our paper, as well as the important guiding significance to our researches. In the following, we have provided detailed responses to each of the reviewers' comments.

Please ensure that your manuscript meets PLOS ONE's style requirements, including those for file naming. The PLOS ONE style templates can be found at https://journals.plos.org/plosone/s/file?id=wjVg/PLOSOne_formatting_sample_main_body.pdf and https://journals.plos.org/plosone/s/file?id=ba62/PLOSOne_formatting_sample_title_authors_affiliations.pdf

Thank you for your reminder and suggestion. For reasons of data privacy, we are sorry that we cannot publish our data directly on a data disclosure platform. However, we have indicated in the text that the data will be made available to the corresponding author upon reasonable request.

Thank you for your suggestion. However, We're sorry that due to the restrictions of the research ethics regulations of the first author's supervising unit (Tongji University), the data related to this study cannot be fully disclosed by uploading it to a public platform. However, the data will be made available upon reasonable request to the corresponding author.

Additional Editor Comments:

The manuscript has a good deal of merit for publication after major revisions. The recommended citation is optional and should only be taken as a suggestion for the improvement of the manuscript.

Reviewers' comments:

Reviewer's Responses to Questions

Comments to the Author

1. Is the manuscript technically sound, and do the data support the conclusions?

Reviewer #1: Partly

2. Has the statistical analysis been performed appropriately and rigorously?

Reviewer #1: Yes

3. Have the authors made all data underlying the findings in their manuscript fully available?

Reviewer #1: No

4. Is the manuscript presented in an intelligible fashion and written in standard English?

Reviewer #1: Yes

5. Review Comments to the Author

Reviewer #1: I am pleased to have the opportunity to review your manuscript. The research topic is meaningful, but the manuscript currently has numerous issues that need to be addressed. In its current state, it is not sufficient for publication in this journal. Below are my detailed comments and suggestions for improvement:

Abstract

Q1: The abstract lacks a clear statement of the research gap.

Thank you for your comment. Our previous abstract did not clearly explain the gap between this study and previous studies. We have now added this to the objectives section. Due to the length limit for the abstract section, we will provide a more detailed introduction to the research gap between this study and previous studies in the introduction.

"Self-efficacy and mindset have been found to moderate the effect of negative life events on individuals' anxiety and depression levels. However, few studies have extended this conclusion to postgraduate student samples. "

Q2: The conclusion should include both theoretical and practical implications of all findings (e.g., the significance of Research Self-efficacy).

Thank you for your suggestion. We have now added a conclusion to the abstract, emphasizing the importance of the study's results for the promotion of the theory of self-determination in the field of education.

"In addition, this study extends the application of self-determination theory to the field of education by verifying the mediating role of research self-efficacy."

Q3: In conclusion, the author expanded the scope of the research results. Anxiety is only one aspect of mental health, and this study is a cross-sectional quantitative study. The measurement of "anxiety" and "mental health" differs in scope and tools. Therefore, the conclusions should be carefully revised to align with the specific dependent variable (anxiety) studied.

Thank you for your suggestion, and we have unified the point you have pointed out. It is inappropriate to generalize the results of anxiety and depression to mental health. Therefore, we have replaced “mental health” in the summary with the more accurate “anxiety and depression” to avoid confusion.

Introduction

Q4: While the authors clearly explain why this study was conducted, the introduction lacks an analysis of the research gap and the study’s innovation.

Thank you for your suggestion, and we have made further revisions to the introduction. We cite several research articles published within the past five years and compare the research gap between this study and these studies, including poor generalizability due to the sample scope and insufficient exploration of moderating effects.

Q5: Before discussing the relationships between variables, the author lacks the reason and basis for studying the mediator and moderator. Specifically, what is the necessity of studying "Research Self-efficacy" and "Mindset"?

Thank you for your comment. The reason we chose to study the Research Self-efficacy variable was that previous studies have found that Research Self-efficacy has a mediating effect in this topic, but the sample size limited the generalizability of the previous research. At the same time, Mindset has been found to significantly moderate depressive symptoms caused by subjective social class and other factors, and it has been widely used in educational psychology. Therefore, this study explored whether Mindset can moderate depressive and anxiety symptoms caused by the student-teacher relationship. We have made further revisions to the introductory section.

Q6: The literature review should not merely list studies or evidence but should critically analyze the literature. Without this, the innovation and necessity of the study are questionable.

Thank you for your suggestion. We have added some content to provide a more detailed introduction to the more relevant literature, including the research sample and detailed conclusions, as well as an analysis of the shortcomings of previous studies.

Q7: The literature review on the "The moderating role of mindset" has critical flaws, leading to logical confusion. First, the current literature reasoning does not sufficiently support H3 and H4; the reasoning suggests that "mindset" might function more as a mediator. Second, "mindset" and "growth mindset" are distinct concepts and variables. The authors should confirm and unify the variables and hypotheses (ensuring consistency in the title, hypotheses, and measurement tools). I recommend referencing high-quality core literature in the field as a logical paradigm.

Thank you for your suggestion. We have now improved this section. We cite and describe in detail the research conducted in recent years on the moderating effect of mindset in mental health to strengthen the basis of the hypothesis. At the same time, regarding the relationship between mindset and growth mindset, we cite Carol's research at the beginning of this section and explain that Carol distinguishes between a fixed mindset and a growth mindset to avoid possible confusion.

Q8: The literature review lacks sufficient support from recent studies (within the last five years).

Thank you for your suggestion. We admit that we have previously cited too few recent studies. In this revision, we have added more studies from the past five years and discussed them in more detail.

Research Methods

Q9: Please provide a detailed description of the sampling process. For example, how were the questionnaires distributed? How was bias controlled during the distribution process? What was the number of valid questionnaire? What were the inclusion and exclusion criteria for valid questionnaire? Was the minimum sample size sufficient, and what is the theoretical basis for it?

For guidance on bias control, please refer to: Podsakoff, P. M., MacKenzie, S. B., Lee, J. Y., & Podsakoff, N. P. (2003). Common method biases in behavioral research: a critical review of the literature and recommended remedies. Journal of Applied Psychology, 88(5), 879. https://doi.org/10.1037/0021-9010.88.5.879 (This is optional should only be taken as a suggestion for the improvement of the manuscript).

Thank you for your question about common method biases. In this revision, we enhanced the description of the sampling process and bias control measures to provide a more comprehensive response to the editor’s suggestions. First, we clarified the specific channels through which the survey was distributed, including email and social media. We also elaborated on the screening and bias control criteria, such as excluding duplicate responses, questionnaires with abnormal completion times, and those with more than one-third missing data. Additionally, we emphasized that the survey was conducted anonymously to minimize common method bias, referencing the recommendations of Podsakoff et al. (2003) on reducing biases related to social desirability. Finally, we acknowledged the limitations of relying solely on these methods to eliminate common method bias and discussed future suggestions for further reduction of such biases.

"The questionnaire was collected using the Wenjuanxing platform, participants were invited to complete the survey through electronic links distributed via email and social media. Responses that were incomplete or exhibited patterns of response bias (e.g., completion times under 5 minutes or over 10 minutes) were excluded from analysis. Only fully completed questionnaires with valid data were retained, and all responses recorded were used for analysis. Responses were collected anonymously to reduce common method bias (Podsakoff et al., 2003)."

Q10: Regarding the measurement tools, I have concerns about the "Mentoring Functions Questionnaire-9 (MFQ-9)" and the "Mindset Scale." Can these tools accurately measure "supervisor-student relationship" and "growth mindset"? Additionally, what is the reliability of these tools?

Thank you for your suggestions.These two scales have been validated as valid instruments in other studies. For example, Hu et al. (2014) measured the quality of supervisor-student relationships among Taiwanese students using the MFQ-9 and obtained a reliability of 0.93; in addition to this, Midkiff et al. (2014) analyzed the mindset scale using Item Response Theory (IRT), with a Cronbach's α = 0.93.Therefore, we believe that these two scales are appropriate measures of the mentor-student relationship as well as the growth mindset. We have used these data to update the relevant indicators in the article and have indicated the cited literature.

Q11: The section on ethical approval should be moved before the description of the “Participants and settings”.

Thank you for your suggestion, we have moved this section to the appropriate section.

Results

Q12: How many questionnaires were used for the confirmatory factor analysis (CFA)? Note that, methodologically, CFA should not use the same data as exploratory factor analysis (i.e., pilot study).

Thank you for your analysis. In this study, we first recruited a group of participants (n = 325) to complete the pilot study and used these data to perform a CFA. After verifying the reliability of the scale, we recruited a larger sample of participants (n = 2,278) to complete the data collection and analysis in the formal study. Therefore, the two parts did not use the same participants.

Q13: In addition to model fit indices (Table 1), what are the other structural validity indicators (e.g., convergent validity, discriminant validity)?

Thank you for your suggestions. We did not report all indicators in full in our initial manuscript submission. In the current version, we first reported in the text that all items had factor loadings > 0.7, and computed Average Variance Extracted (AVE) and Composite Reliability (CR) values for each scale from the factor loadings in each scale and added them to Table 1. AVE and CR are indicators of the consistency of the question items within the scale. All of the scales used in this study have a CR > 0.7 and an AVE > 0.5, which provides good reliability. For discriminant validity, we calculated the heterotrait-monotrait ratio (HTMT) between the scales and added it to Table 2. The results showed that the HTMT between all the scales was < 0.85, indicating good discriminant validity between the scales.

Q14: Please provide literature support for the standard value ranges of all indicators.

Thank you for your advice. We have detailed the range and significance of each indicator in the footnotes to Table 1, citing the

---

## [Decision Letter · Decision Letter 1]

26 Jun 2025

How do Supervisor-Student Relationship Affect Anxiety of Graduate Students?

——The Mediating Role of Research Self-efficacy and the Moderating role of Mindset

PONE-D-25-05351R1

Dear Dr. Songjie Huang,

We’re pleased to inform you that your manuscript has been judged scientifically suitable for publication and will be formally accepted for publication once it meets all outstanding technical requirements.

Kind regards,

Naeem Mubarak, PhD

Academic Editor

PLOS ONE

Additional Editor Comments (optional):

The authors have addressed all the comments. No further changes required

Reviewers' comments:

Reviewer's Responses to Questions

**Comments to the Author**

Reviewer #1: All comments have been addressed

2. Is the manuscript technically sound, and do the data support the conclusions?

Reviewer #1: Yes

3. Has the statistical analysis been performed appropriately and rigorously?

Reviewer #1: Yes

4. Have the authors made all data underlying the findings in their manuscript fully available?

Reviewer #1: Yes

5. Is the manuscript presented in an intelligible fashion and written in standard English?

Reviewer #1: Yes

Reviewer #1: Many thanks to the authors for their efforts in revising the manuscript. Although there are still some issues that need to be strengthened in the research methods section (for example, the theoretical basis for the minimum sample size, etc.), the overall quality has been improved. I recommend the acceptance of this manuscript for publication.

**Do you want your identity to be public for this peer review?** For information about this choice, including consent withdrawal, please see our Privacy Policy

Reviewer #1: No

---

## [Editor Report · Acceptance letter]

PONE-D-25-05351R1

PLOS ONE

Dear Dr. Huang,

I'm pleased to inform you that your manuscript has been deemed suitable for publication in PLOS ONE. Congratulations! Your manuscript is now being handed over to our production team.

Kind regards,

on behalf of

Dr Naeem Mubarak

Academic Editor

PLOS ONE